

# Tropospheric water vapour isotopologue data ($H_2^{16}O$, $H_2^{18}O$ and $HD^{16}O$) as obtained from NDACC/FTIR solar absorption spectra

Sabine Barthlott[1], Matthias Schneider[1], Frank Hase[1], Thomas Blumenstock[1], Matthäus Kiel[1], Darko Dubravica[1], Omaira E. García[2], Eliezer Sepúlveda[2], Gizaw Mengistu Tsidu[3,4], Samuel Takele Kenea[3,*], Michel Grutter[5], Eddy F. Plaza[5], Wolfgang Stremme[5], Kim Strong[6], Dan Weaver[6], Mathias Palm[7], Thorsten Warneke[7], Justus Notholt[7], Emmanuel Mahieu[8], Christian Servais[8], Nicholas Jones[9], David W. T. Griffith[9], Dan Smale[10], and John Robinson[10]

[1]Institute of Meteorology and Climate Research (IMK-ASF), Karlsruhe Institute of Technology (KIT), Karlsruhe, Germany
[2]Izaña Atmospheric Research Center, Agencia Estatal de Meteorología (AEMET), Santa Cruz de Tenerife, Spain
[3]Department of Physics, Addis Ababa University, P.O. Box 1176, Addis Ababa, Ethiopia
[4]Botswana International University of Technology and Science (BIUST) Priv. Bag 16, Palapye, Botswana
[5]Centro de Ciencias de la Atmósfera, Universidad Nacional Autónoma de México, 04510 Ciudad de México, México
[6]Department of Physics, University of Toronto, Toronto, Ontario, Canada
[7]Institute of Environmental Physics, University of Bremen, Bremen, Germany
[8]Institute of Astrophysics and Geophysics, University of Liège, Liège, Belgium
[9]Centre for Atmospheric Chemistry, University of Wollongong, Wollongong, New South Wales, Australia
[10]National Institute of Water and Atmospheric Research, Lauder, New Zealand
[*]now at: Department of Physics, Samara University, P.O. BOX 132, Samara, Ethiopia

*Correspondence to:* S. Barthlott
(sabine.barthlott@kit.edu)

**Abstract.** We report on the ground-based FTIR (Fourier Transform InfraRed) tropospheric water vapour isotopologue remote sensing data that have been recently made available via the database of NDACC (Network for the Detection of Atmospheric Composition Change; ftp://ftp.cpc.ncep.noaa.gov/ndacc/MUSICA/) and via doi:10.5281/zenodo.48902. Currently, data are available for 12 globally distributed stations. They have been centrally retrieved and quality filtered in the framework of the MUSICA project (MUlti-platform remote Sensing of Isotopologues for investigating the Cycle of Atmospheric water). We explain particularities of retrieving the water vapour isotopologue state (vertical distribution of $H_2^{16}O$, $H_2^{18}O$, and $HD^{16}O$) and reveal the need for a new metadata template for archiving FTIR isotopologue data. We describe the format of different data components and give recommendations for correct data usage. Data are provided as two data types. The first type is best-suited for tropospheric water vapour distribution studies disregarding different isotopologues (comparison with radiosonde data, analyses of water vapour variability and trends, etc.). The second type is needed for analysing moisture pathways by means of {$H_2O,\delta D$}-pair distributions.

## 1 Introduction

Simultaneous observations of different tropospheric water isotopologues can provide valuable information on moisture source, transport, cloud processes, and precipitation (e.g. Dansgaard, 1964; Gat, 2000; Yoshimura et al., 2004). For identifying





and analysing different tropospheric moisture pathways, the distribution of {$H_2O$,$\delta D$}-pairs (e.g. Galewsky et al., 2005; Noone, 2012; González et al., 2015) or deuterium-excess ($d = \delta D - 8\delta^{18}$, e.g. Pfahl and Sodemann, 2014; Steen-Larsen et al., 2014; Aemisegger et al., 2014) is particularly promising. The $\delta$-notation is used to express the relation of the observed isotopologue ratio to the standard ratio VSMOW (Vienna Standard Mean Ocean Water), whereby $\delta D = \frac{HD^{16}O/H_2^{16}O}{VSMOW} - 1$ and $\delta^{18} = \frac{H_2^{18}O/H_2^{16}O}{VSMOW} - 1$.

In recent years, there has been significant progress in measuring the tropospheric water vapour isotopologues; remote sensing observations are particularly interesting since they can provide data for the free troposphere and they can be performed continuously (for cloud-free conditions). During MUSICA (Multi-platform remote Sensing of Isotopologues for investigating the Cycle of Atmospheric water), a method has been developed to obtain tropospheric water vapour profiles as well as {$H_2O$,$\delta D$}-pairs from ground-based FTIR (Fourier Transform InfraRed) and space-based IASI (Infrared Atmospheric Sounding Interferometer) observations. The ground-based FTIR spectra are measured in the framework of the NDACC (Network for the Detection of Atmospheric Composition Chang, www.ndacc.org). These spectra are of very high quality and at some stations available since the 1990s, thus offering long-term data records, which are of particular interest for climatological studies or for assessing the stability of long-term satellite data records. Network-wide consistent quality of the MUSICA NDACC/FTIR water vapour isotopologue data is ensured by a central data processing and quality control (e.g. all data are quality filtered by the $XCO_2$ method as presented in Barthlott et al., 2015). The high quality of the MUSICA ground- and space-based remote sensing data has been demonstrated by empirical validation studies (Schneider et al., 2016).

In this paper, we present the MUSICA NDACC/FTIR data as provided recently via the NDACC database (see also Barthlott et al., 2016). Our objective is to make a data user aware of the nature of the data and to give recommendations and explanations for correct data usage. Section 2 briefly describes the particularity of retrievals of the water vapour isotopologue state. It is shown that for achieving an optimal estimation of the tropospheric water vapour isotopologue state, we have to work with full state vectors consisting of different isotopologues, i.e. a state vector consisting of several trace gases. As a result, water vapour isotopologue data can not be provided in exactly the same data format as other FTIR data. In Sect. 3 we describe the data format used for providing this new data via the NDACC database. Section 4 gives some insight into the data characteristics and recommendation for working with this new dataset. In Section 5, we conclude and summarize our results. The summarized guidelines for a correct data usage can be found in Table 3.

## 2 Optimal estimation of the water vapour isotopologue state

### 2.1 The principle of optimal estimation retrieval methods

Atmospheric remote sensing retrievals characterize an atmospheric state from a measured spectrum. However, such an inversion problem is often ill-posed (a lot of different atmospheric states can explain the measured spectrum). Consequently for solving this problem, some kind of regularisation is required. This can be introduced by means of a cost function:

$$[\boldsymbol{y} - \boldsymbol{F}(\boldsymbol{x},\boldsymbol{p})]^T \mathbf{S}_\epsilon^{-1}[\boldsymbol{y} - \boldsymbol{F}(\boldsymbol{x},\boldsymbol{p})] + [\boldsymbol{x} - \boldsymbol{x}_a]^T \mathbf{S_a}^{-1}[\boldsymbol{x} - \boldsymbol{x}_a]. \tag{1}$$





Here, the first term is a measure of the difference between the measured spectrum ($\boldsymbol{y}$) and the spectrum simulated for a given atmospheric state ($\boldsymbol{x}$), where $\boldsymbol{F}$ represents the forward model, which simulates a spectra $\boldsymbol{y}$ for a given state $\boldsymbol{x}$, taking into account the actual measurement noise level ($\mathbf{S}_\epsilon$ is the measurement noise covariance). Vector $\boldsymbol{p}$ represents auxiliary atmospheric parameters (like temperature) or instrumental characteristics (like the instrumental line shape). The second term

of the cost function (1) is the regularisation term. It constrains the atmospheric solution state ($\boldsymbol{x}$) towards an a priori most likely state ($\boldsymbol{x_a}$), whereby kind and strength of the constraint are defined by the a priori covariance matrix $\mathbf{S_a}$. The constrained solution is reached at the minimum of the cost function (1). This method of updating the knowledge about the a priori state with information from a measurement is known as the optimal estimation method which is a standard remote sensing retrieval method (see Rodgers, 2000, for more details), whereby validity of the optimal estimation solution strongly depends on a correct

and comprehensive description of the a priori state by means of $\boldsymbol{x_a}$ and $\mathbf{S_a}$.

## 2.2  Correct description of the water vapour isotopologue state

Water vapour isotopologues with reasonably strong and well discernible spectral infrared signatures are $H_2^{16}O$, $H_2^{18}O$, and $HD^{16}O$. Hereafter, we will refer to the combined states of $H_2^{16}O$, $H_2^{18}O$ and $HD^{16}O$ as the full water vapour isotopologue state.

Tropospheric water vapour shows a strong variation (in space and time) and it can be better described by a log-normal than

by a normal distribution (Hase et al., 2004; Schneider et al., 2006). Furthermore, the different isotopologues vary mostly in parallel, i.e. their variations are strongly correlated and the variation of the isotopologue ratios is much smaller. For instance, the variation of $\ln[HD^{16}O] - \ln[H_2^{16}O]$ is about one order of magnitude smaller and the variations of $\ln[H_2^{18}O] - \ln[H_2^{16}O]$ is about two orders of magnitude smaller than the correlated variations of $\ln[H_2^{16}O]$, $\ln[H_2^{18}O]$ and $\ln[HD^{16}O]$. An elegant method for correctly describing the nature of the full water vapour isotopologue state is to work on a logarithmic scale (due to

the log-normal distribution characteristic) and with the following three states:

the humidity-proxy-state:

$$\frac{1}{3}(\ln[H_2^{16}O] + \ln[H_2^{18}O] + \ln[HD^{16}O]), \tag{2}$$

the $\delta$D-proxy-state:

$$\ln[HD^{16}O] - \ln[H_2^{16}O], \tag{3}$$

the deuterium-excess-proxy-state (d-proxy-state):

$$\ln[HD^{16}O] - \ln[H_2^{16}O] - 8(\ln[H_2^{18}O] - \ln[H_2^{16}O]). \tag{4}$$

The water vapour isotopologue state can be expressed in the basis of $\{\ln[H_2^{16}O], \ln[H_2^{18}O], \ln[HD^{16}O]\}$ or in the basis of the proxies of $\{\text{humidity}, \delta D, d\}$. Both expressions are equivalent. Each basis has the dimension ($nol \times 3$), whereby $nol$ is the number of levels of the radiative transfer model atmosphere and three different combinations of the three different isotopologues have to be considered. In the following, the full water vapour isotopologue state vector expressed in the

$\{\ln[H_2^{16}O], \ln[H_2^{18}O], \ln[HD^{16}O]\}$-basis and the $\{\text{humidity}, \delta D, d\}$-proxy basis will be referred to as $\boldsymbol{x^l}$ and $\boldsymbol{x^{l\prime}}$, respectively.





A basis transformation can be achieved by operator $\mathbf{P}$:

$$\mathbf{P} = \begin{pmatrix} \frac{1}{3}\mathbb{I} & \frac{1}{3}\mathbb{I} & \frac{1}{3}\mathbb{I} \\ -\mathbb{I} & 0 & \mathbb{I} \\ 7\mathbb{I} & -8\mathbb{I} & \mathbb{I} \end{pmatrix}. \tag{5}$$

Here, the nine matrix blocks have the dimension $(nol \times nol)$, $\mathbb{I}$ stands for an identity matrix and the state vectors $x^l$ and $x^{l'}$ are related by:

$$x^{l'} = \mathbf{P}x^l. \tag{6}$$

Similarly, covariance matrices can be expressed in the two basis systems and the respective matrices $\mathbf{Sa^l}$ and $\mathbf{Sa^{l'}}$ are related by:

$$\mathbf{S^{l'}} = \mathbf{P}\mathbf{S^l}\mathbf{P}^T. \tag{7}$$

The variation in humidity, $\delta$D, and deuterium excess have different magnitudes. This can be well considered by giving the a priori covariance matrix in the $\{\mathrm{humidity}, \delta\mathrm{D}, \mathrm{d}\}$-proxy basis:

$$\mathbf{Sa^{l'}} = \begin{pmatrix} \mathbf{Sa^l_{hum}} & 0 & 0 \\ 0 & \mathbf{Sa^l_{\delta D}} & 0 \\ 0 & 0 & \mathbf{Sa^l_d} \end{pmatrix}. \tag{8}$$

This assumes that humidity, $\delta$D and deuterium excess vary independently, which is actually not the case in the lower/middle troposphere. However, the correlation between humidity, $\delta$D, and deuterium excess is a detail. It is important that the different magnitudes of variability are taken into account (the entries of $\mathbf{Sa^l_{hum}}$ are much larger than the entries of $\mathbf{Sa^l_{\delta D}}$ and $\mathbf{Sa^l_d}$).

The covariance matrix $\mathbf{Sa^l}$ can be calculated as:

$$\mathbf{Sa^l} = \mathbf{P}^{-1}\mathbf{Sa^{l'}}\mathbf{P}^{-T}$$

$$= \begin{pmatrix} \mathbb{I} & -\frac{3}{8}\mathbb{I} & \frac{1}{24}\mathbb{I} \\ \mathbb{I} & -\frac{1}{4}\mathbb{I} & -\frac{1}{12}\mathbb{I} \\ \mathbb{I} & \frac{5}{8}\mathbb{I} & \frac{1}{24}\mathbb{I} \end{pmatrix} \begin{pmatrix} \mathbf{Sa^l_{hum}} & 0 & 0 \\ 0 & \mathbf{Sa^l_{\delta D}} & 0 \\ 0 & 0 & \mathbf{Sa^l_d} \end{pmatrix} \begin{pmatrix} \mathbb{I} & \mathbb{I} & \mathbb{I} \\ -\frac{3}{8}\mathbb{I} & -\frac{1}{4}\mathbb{I} & \frac{5}{8}\mathbb{I} \\ \frac{1}{24}\mathbb{I} & -\frac{1}{12}\mathbb{I} & \frac{1}{24}\mathbb{I} \end{pmatrix}$$

$$= \begin{pmatrix} \mathbf{Sa^l_{hum}} + \frac{9}{8^2}\mathbf{Sa^l_{\delta D}} + \frac{1}{24^2}\mathbf{Sa^l_d} & \mathbf{Sa^l_{hum}} + \frac{6}{8^2}\mathbf{Sa^l_{\delta D}} - \frac{2}{24^2}\mathbf{Sa^l_d} & \mathbf{Sa^l_{hum}} - \frac{15}{8^2}\mathbf{Sa^l_{\delta D}} + \frac{1}{24^2}\mathbf{Sa^l_d} \\ \mathbf{Sa^l_{hum}} + \frac{6}{8^2}\mathbf{Sa^l_{\delta D}} - \frac{2}{24^2}\mathbf{Sa^l_d} & \mathbf{Sa^l_{hum}} + \frac{4}{8^2}\mathbf{Sa^l_{\delta D}} + \frac{4}{24^2}\mathbf{Sa^l_d} & \mathbf{Sa^l_{hum}} - \frac{10}{8^2}\mathbf{Sa^l_{\delta D}} - \frac{2}{24^2}\mathbf{Sa^l_d} \\ \mathbf{Sa^l_{hum}} - \frac{15}{8^2}\mathbf{Sa^l_{\delta D}} + \frac{1}{24^2}\mathbf{Sa^l_d} & \mathbf{Sa^l_{hum}} - \frac{10}{8^2}\mathbf{Sa^l_{\delta D}} - \frac{2}{24^2}\mathbf{Sa^l_d} & \mathbf{Sa^l_{hum}} + \frac{25}{8^2}\mathbf{Sa^l_{\delta D}} + \frac{1}{24^2}\mathbf{Sa^l_d} \end{pmatrix}. \tag{9}$$

This matrix $\mathbf{Sa^l}$ correctly captures the covariance of the a priori state in the $\{\ln[\mathrm{H_2^{16}O}], \ln[\mathrm{H_2^{18}O}], \ln[\mathrm{HD^{16}O}]\}$-basis and it is important for a correct formulation of the cost function (1) and thus for setting up a correctly constrained optimal estimation retrieval of the water vapour isotopologue state.

Matrix $\mathbf{Sa^l}$ reveals the complex covariances between the state vector components of $x^l$, whereby it has to be considered that the entries of $\mathbf{Sa^l_{hum}}$ are much larger than the entries of $\mathbf{Sa^l_{\delta D}}$ and significantly larger than the entries of $\mathbf{Sa^l_d}$, which means





that the nine ($nol \times nol$) blocks of $\mathbf{Sa^l}$ have almost the same entries. The variations in $\ln[\mathrm{H}_2^{16}\mathrm{O}]$, $\ln[\mathrm{H}_2^{18}\mathrm{O}]$, and $\ln[\mathrm{HD}^{16}\mathrm{O}]$ are strongly correlated and we cannot work with individual state vectors that report the states of the different isotopologues independently. Instead, we have to describe the full isotopologue state by a single state vector with the dimension ($nol \times 3$). This is the reason why MUSICA NDACC/FTIR water vapour isotopologue data cannot be provided in the same data format as

other NDACC/FTIR data. A slightly extended data format is needed, which is described in the next section.

## 2.3   The MUSICA (v2015) ground-based NDACC/FTIR retrieval setup

In Schneider et al. (2012), the MUSICA NDACC/FTIR retrieval setup, like interfering gases, temperature fit, etc., is described in great detail. Here, we focus on the modification made for the retrieval version (v2015).

For the previous retrieval version, we used 11 spectral windows with lines of water vapour isotopologues (see Fig. 2 of

Schneider et al., 2012). For the final MUSICA retrieval version (v2015), we removed windows with strong $\mathrm{H}_2^{16}\mathrm{O}$ lines and replaced them by windows with weaker lines. By this modification, we want to make sure that even for very humid sites observed spectral lines do not saturate. Furthermore, we add a second window where a $\mathrm{H}_2^{18}\mathrm{O}$ signature dominates. The nine spectral water vapour isotopologue windows are depicted in Fig. 1 for an observation and fit, which are typical for the different NDACC stations. In addition, we fit the three spectral windows with the $\mathrm{CO}_2$ lines, which is beneficial for atmospheric temperature re-

trievals (the $\mathrm{CO}_2$ lines are the same as for the previous retrieval version: between $2610.35\text{-}2610.8\,\mathrm{cm}^{-1}$, $2613.7\text{-}2615.4\,\mathrm{cm}^{-1}$ and $2626.3\text{-}2627.0\,\mathrm{cm}^{-1}$; not shown). All these spectral windows are covered by NDACC filter #3.

For v2015 we perform an optimal estimation of the $\{\mathrm{humidity}, \delta\mathrm{D}, \mathrm{d}\}$-proxy states as explained in Sect. 2.2. This is a further development of the previous retrieval version (there the optimal estimation was made for the $\{\mathrm{humidity}, \delta\mathrm{D}\}$-proxy states, Sects. 3 and 4 of Schneider et al., 2012). Consistent with the previous version, we perform simultaneous but individual

fits (no cross-constraints) for profiles of the water vapour isotopologue $\mathrm{H}_2^{17}\mathrm{O}$, temperature, and interfering species $\mathrm{CO}_2$, $\mathrm{O}_3$, $\mathrm{N}_2\mathrm{O}$, $\mathrm{CH}_4$, and HCl.

For the previous retrieval version, we used HITRAN 2008 spectroscopic water vapour line parameters (Rothman et al., 2009) and adopted it for speed-dependent Voigt line shapes (Schneider et al., 2011). For v2015, we work with HITRAN 2012 parameters (Rothman et al., 2013) and optimised it for speed-dependent Voigt line parameterisation. For further optimisation

of the HITRAN parameters, we used high quality $\mathrm{H}_2\mathrm{O}$ and $\delta\mathrm{D}$ aircraft in-situ profile references, coincident FTIR spectra, and FTIR spectra measured at three rather distinct sites (Izaña on Tenerife Island, subtropical ocean, Karlsruhe, central Europe, and Kiruna, northern Sweden). This method for line parameter optimisation by means of atmospheric spectra is described in detail in Schneider and Hase (2009) and Schneider et al. (2011). We changed line intensities and broadening parameters by about 5-10%, which is in agreement with the uncertainty values as given in the HITRAN parameter files. More details on the

modification of the line parameters as shown in Fig. 1 are given in the Appendix A.

The empirical assessment study of Schneider et al. (2016) suggests an accuracy for the MUSICA (v2015) NDACC/FTIR $\mathrm{H}_2\mathrm{O}$ and $\delta\mathrm{D}$ products of about 10% and 10‰, respectively.



## 3   Dataset description

MUSICA NDACC/FTIR water vapour isotopologue data are available via the NDACC database and can be accessed via ftp://ftp.cpc.ncep.noaa.gov/ndacc/MUSICA/ and via doi:10.5281/zenodo.48902. The data are provided as HDF4 files and they have been generated in compliance with GEOMS (Generic Earth Observation Metadata Standard).

For isotopologue data, a new metadata template GEOMS-TE-FTIR-ISO-001 has been set up. It is almost identical with the GEOMS-TE-FTIR template (used for all other FTIR data provided via NDACC in HDF4 format), but with the additional variable "CROSSCORRELATE.N". For MUSICA data, this new variable has three entries "$H_2^{16}O$", "$H_2^{18}O$", and "$HD^{16}O$". This enables the three trace gases $H_2^{16}O$, $H_2^{18}O$, and $HD^{16}O$ to be provided as one full state vector (vector with $(nol \times 3)$ elements) together with their full averaging kernels and full error covariances (both of $((nol \times 3) \times (nol \times 3))$ dimension) in

one data file. To be compliant with GEOMS, the data are provided on a linear scale (and not on a logarithmic scale on which the inversion is performed, see Sect. 2).

All the isotopologue data have been normalized with respect to their natural isotopologue abundances. These natural abundances are 0.997317 for $H_2^{16}O$, 0.002000 for $H_2^{18}O$, and $3.106930 \times 10^{-4}$ for $HD^{16}O$, respectively. This normalization means that $\delta D$ can directly be calculated as $(\frac{[HD^{16}O]}{[H_2^{16}O]} - 1)$ from provided $H_2^{16}O$, $H_2^{18}O$ and $HD^{16}O$ amounts. Similarly, deuterium-

excess can be calculated as $(\frac{[HD^{16}O]}{[H_2^{16}O]} - 1) - 8 \times (\frac{[H_2^{18}O]}{[H_2^{16}O]} - 1)$.

Two different data types are provided. The first type is stored in HDF files called "ftir.iso.h2o". These files report the best estimate of the $H_2^{16}O$ state, but its usefulness for isotopologue studies (e.g. in the context of $\{H_2O, \delta D\}$-distribution plots) is significantly compromised. The second type is stored in HDF files called "ftir.iso.post.h2o". This data type should be used for analysing $\{H_2O, \delta D\}$-distribution plots. Section 4 gives more details on the different data types.

The number of stations contributing to the MUSICA NDACC/FTIR dataset is gradually increasing and Table 1 gives an overview of the current status (status for January 2016) together with mean DOFS (degrees of freedom of signal) for the two data types. The stations are well distributed from the Arctic to the Antarctic and in some occasions offer data since the late 1990s. A further extension of this dataset to other sites or for some stations to measurements made in the beginning of the 1990s is feasible but was not possible within the MUSICA project.

## 3.1   Averaging kernels

An averaging kernel describes how a retrieved state vector responds to variations of the real atmospheric state vector. The MUSICA NDACC/FTIR isotopologue state vector consists of three trace gases and the corresponding full averaging kernel matrix $\mathbf{A}$ consists of nine blocks, each of which is a $(nol \times nol)$ matrix. In total, $\mathbf{A}$ has the dimension $((nol \times 3) \times (nol \times 3))$:

$$\mathbf{A} = \begin{pmatrix} \mathbf{A_{11}} & \mathbf{A_{12}} & \mathbf{A_{13}} \\ \mathbf{A_{21}} & \mathbf{A_{22}} & \mathbf{A_{23}} \\ \mathbf{A_{31}} & \mathbf{A_{32}} & \mathbf{A_{33}} \end{pmatrix}. \tag{10}$$

The three blocks along the diagonal describe the direct responses: block $\mathbf{A_{11}}$ for the response of the $H_2^{16}O$ retrieval product to real atmospheric $H_2^{16}O$ variations, block $\mathbf{A_{22}}$ for the response of the $H_2^{18}O$ retrieval product to real atmospheric $H_2^{18}O$



variations and block $\mathbf{A_{33}}$ for the response of the HD$^{16}$O retrieval product to real atmospheric HD$^{16}$O variations. The outer diagonal blocks describe the cross responses: two blocks for the response of the H$_2^{16}$O retrieval product to real atmospheric H$_2^{18}$O and HD$^{16}$O variations ($\mathbf{A_{12}}$ and $\mathbf{A_{13}}$, respectively), two blocks for the response of the H$_2^{18}$O retrieval product to real atmospheric H$_2^{16}$O and HD$^{16}$O variations ($\mathbf{A_{21}}$ and $\mathbf{A_{23}}$, respectively) and two blocks for the response of the HD$^{16}$O retrieval

product to real atmospheric H$_2^{16}$O and H$_2^{18}$O variations ($\mathbf{A_{31}}$ and $\mathbf{A_{32}}$, respectively). The different blocks of the matrix (see Fig. 2) can be easily identified by the values of the variable "CROSSCORRELATE.N": 1 = H$_2^{16}$O, 2 = H$_2^{18}$O, and 3 = HD$^{16}$O.

### 3.2  Error covariances

For error calculations, same uncertainty sources are assumed for all MUSICA NDACC/FTIR stations and are grouped into statistical and systematic errors. An overview of the uncertainty assumptions is given in Table 2. Error covariances for each

of these uncertainty sources are calculated according to Rodgers (2000). Total error covariances are obtained as the sum of the individual error covariances, calculated separately for statistical and systematic errors. Full error covariance matrices have the same matrix dimension as the full averaging kernel matrix and provide information about the errors and how the errors between different altitudes as well as between the three different isotopologues are correlated.

### 3.3  Column densities

Partial and total column densities are also provided as well as column sensitivity averaging kernels and column error covariances. Partial columns are calculated for the layers between the $nol$ atmospheric levels. A full water vapour isotopologue partial column state vector has $((nol - 1) \times 3)$ elements (i.e., $(nol - 1)$ elements for each of the three isotopologues). The column sensitivity averaging kernel has the dimension $(((nol - 1) \times 3) \times 3)$ and describes the sensitivities of the total column retrieval product (columns of H$_2^{16}$O, H$_2^{18}$O and HD$^{16}$O) with respect to real atmospheric variations in the isotopologues' partial

columns. Column error covariances are provided for statistical and systematic errors independently and are matrices with the dimension $(3 \times 3)$. They describe total column errors for the three isotopologues and how these errors are correlated.

All column data are calculated from mixing ratio data (mixing ratio state vector, error covariances and averaging kernels). For a conversion of mixing ratios (ppmv) to number densities (molec./cm$^3$), we use the ideal gas law and work with retrieved temperature profiles. Temperature and pressure profiles are also provided in the HDF files.

Please note: Since GEOMS requires the same number of elements of column data and the respective mixing ratio data, value 0 has been added for all colum data values at level $nol$.

## 4  Recommendations for data usage

In order to be compliant with GEOMS, data are provided for the full state vector consisting of the H$_2^{16}$O, H$_2^{18}$O, and HD$^{16}$O state vectors and in mixing ratios (ppmv). However, actually the MUSICA NDACC/FTIR retrieval works on a logarithmic

scale and performs an optimal estimation of combined isotopologue states (see Sect. 2). These details have to be considered when working with these data.



## 4.1 Transfer to a logarithmic scale

For operations with averaging kernels (e.g. when adjusting model data to the sensitivity of a remote sensing system), it has to be considered that the retrieval works on a logarithmic scale, because only on this scale linearity assumptions are valid. Therefore, it is strongly recommended to transfer the averaging kernels to a logarithmic scale, whereby it has to be considered that the derivatives are calculated for the state as given by the retrieval state vector $x$. The full retrieval state vector consists of the retrieved $H_2^{16}O$, $H_2^{18}O$, and $HD^{16}O$ states, i.e. it is a vector with $(nol \times 3)$ elements $x_i$ (with $i$ between 1 and $nol \times 3$). The full averaging kernel matrix has the dimension $(nol \times 3) \times (nol \times 3)$. Figure 2 plots the full averaging kernel matrix provided on a linear scale ($\mathbf{A}$). It has the entries $a_{i,j}$ ($i$ and $j$ are between 1 and $nol \times 3$, whereby $i$ is the row- and $j$ the column-index). Entry $a_{i,j}$ is a derivative, describing how the retrieved state vector element $i$ responses to a variation of the real atmospheric state vector element $j$. Since $\partial \ln x = \frac{\partial x}{x}$, entry $a_{i,j}$ has to be modified for operations on a logarithmic scale as follows:

$$a_{i,j}^l = a_{i,j} \frac{x_j}{x_i}. \tag{11}$$

Here, $a_{i,j}^l$ are the entries of the full averaging kernel matrix transferred to a logarithmic scale ($\mathbf{A^l}$), which is plotted in Fig. 3. This matrix $\mathbf{A^l}$ is the averaging kernel matrix for a scale on which the linearity assumptions are valid and on which kernel operations should be performed.

For the purpose of error analyses, it is also very useful to transfer the error covariances on a logarithmic scale. On this scale, the $\delta D$ and deuterium-excess error covariances can be made available in an elegant manner (see Sect. 4.4). The covariance matrix entry $s_{i,j}$ is the covariance between the state vector elements $i$ and $j$ (the value of these elements is $x_i$ and $x_j$, respectively). On a logarithmic scale, the covariance matrix entry is:

$$s_{i,j}^l = \frac{s_{i,j}}{x_i x_j}. \tag{12}$$

The covariance matrices on a logarithmic scale ($\mathbf{S^l}$) are composed of the entries $s_{i,j}^l$.

## 4.2 Comparison studies

For many purposes, remote sensing data may be compared to other data. For instance, they may be compared to vertically resolved observational data in order to empirically assess the quality of the different datasets, or they are compared to model data in order to investigate model performances. For such comparisons, it is important to consider the sensitivity of the remote sensing system. While highly resolved profile data or model data generally capture atmospheric signals well even on rather small scales, remote sensing data report atmospheric signals according to the averaging kernel. In order to make different datasets comparable, we have to convolve the data with the averaging kernels. A full water vapour isotopologue state vector obtained from highly resolved profile measurements or model calculations ($x_{\mathbf{data}}^l$) will be observed by a remote sensing system as $x_{\mathbf{data,conv}}^l$:

$$x_{\mathbf{data,conv}}^l = \mathbf{A^l}(x_{\mathbf{data}}^l - x_{\mathbf{a}}^l) + x_{\mathbf{a}}^l. \tag{13}$$

Here, $x_{\mathbf{a}}^l$ means the full water vapour isotopologue a priori state vector on a logarithmic scale.



### 4.3 Simple water vapour analyses (neglecting the different isotopologues)

About 99.7% of all water vapour is present in form of the isotopologue $H_2^{16}O$ and for studies in which the different isotopologues are not distinguished, one should work with the retrieved $H_2^{16}O$ data. One example is comparison of the MUSICA NDACC/FTIR remote sensing data with water vapour data measured in-situ by radiosondes. Since radiosondes do not distinguish between isotopologues, they report almost exclusively $H_2^{16}O$ concentrations and amounts.

In order to simulate how a remote sensing system observes a $\ln[H_2^{16}O]$ radiosonde profile, we have to convolve the state vector determined from radiosonde data ($\boldsymbol{x}_{\mathbf{RS}}^l$) using the averaging kernel blocks $\mathbf{A}_{\mathbf{11}}^{\mathbf{l}}$, $\mathbf{A}_{\mathbf{12}}^{\mathbf{l}}$ and $\mathbf{A}_{\mathbf{13}}^{\mathbf{l}}$ (these blocks are depicted in Fig. 3). The radiosonde only measures $\ln[H_2^{16}O]$, it does not measure $\ln[H_2^{18}O]$ and $\ln[HD^{16}O]$. However, $\ln[H_2^{18}O]$ and $\ln[HD^{16}O]$ are strongly correlated to $\ln[H_2^{16}O]$:

$$\ln[H_2^{18}O] - \ln[H_2^{18}O]_a = \ln[H_2^{16}O] - \ln[H_2^{16}O]_a + \frac{1}{8}\{\delta D - \delta D_a - (d - d_a)\} \tag{14}$$

and

$$\ln[HD^{16}O] - \ln[HD^{16}O]_a = \ln[H_2^{16}O] - \ln[H_2^{16}O]_a + \{\delta D - \delta D_a\}. \tag{15}$$

Here, index $a$ identifies a priori data. Like all full isotopologue state vectors, $\boldsymbol{x}_{\mathbf{RS}}^l$ has ($nol \times 3$) components. The first section of $nol$ components is determined by the in-situ measured $\ln[H_2^{16}O]$ profile. The second section of $nol$ components is also determined by the in-situ measured $\ln[H_2^{16}O]$ profile, but we have to consider an uncertainty of these vector components according to the uncertainty covariance matrix $\frac{1}{8^2}(\mathbf{Sa}_{\delta D}^l + \mathbf{Sa}_d^l)$ (see Eq. 14). The uncertainty is due to the fact that a radiosonde does not measure $H_2^{18}O$. The third section of $nol$ components is similarly determined by the in-situ measured $\ln[H_2^{16}O]$ profile, but with an uncertainty according to the uncertainty covariance matrix $\mathbf{Sa}_{\delta D}^l$ (see Eq. 15), whereby the uncertainty is caused by missing $HD^{16}O$ radiosonde measurements. Hence, a $\ln[H_2^{16}O]$ radiosonde profile is equivalent to a $\ln[H_2^{16}O]$ remote sensing observation of

$$(\mathbf{A}_{\mathbf{11}}^{\mathbf{l}} + \mathbf{A}_{\mathbf{12}}^{\mathbf{l}} + \mathbf{A}_{\mathbf{13}}^{\mathbf{l}})(\boldsymbol{x}_{\mathbf{RS}}^l - \boldsymbol{x}_{\boldsymbol{a}}^l) + \boldsymbol{x}_{\boldsymbol{a}}^l, \tag{16}$$

whereby there is an uncertainty in the equivalency of

$$\frac{1}{8^2}\mathbf{A}_{\mathbf{12}}^{\mathbf{l}}(\mathbf{Sa}_{\delta D}^l + \mathbf{Sa}_d^l){\mathbf{A}_{\mathbf{12}}^{\mathbf{l}}}^T + \mathbf{A}_{\mathbf{13}}^{\mathbf{l}}\mathbf{Sa}_{\delta D}^l{\mathbf{A}_{\mathbf{13}}^{\mathbf{l}}}^T, \tag{17}$$

which is due to missing radiosonde observations of $H_2^{18}O$ and $HD^{16}O$.

Equation 16 reveals that the ($nol \times nol$) averaging kernel matrix $\mathbf{A}_{\mathbf{H2O}}^{\mathbf{l}} = \mathbf{A}_{\mathbf{11}}^{\mathbf{l}} + \mathbf{A}_{\mathbf{12}}^{\mathbf{l}} + \mathbf{A}_{\mathbf{13}}^{\mathbf{l}}$ is a good proxy for the remote sensing system's sensitivity with respect to $H_2^{16}O$ and thus water vapour in general (about 99.7% of all water vapour molecules are $H_2^{16}O$ isotopologues). The columns of the matrix $\mathbf{A}_{\mathbf{H2O}}^{\mathbf{l}}$ are plotted in Fig. 4.

### 4.4 Utility of the $\{\mathrm{humidity}, \delta\mathbf{D}, \mathrm{d}\}$-proxy basis and data aposteriori processing

The MUSICA NDACC/FTIR retrieval performs an optimal estimation of the humidity-, $\delta D$-, and deuterium-excess-proxy-states. Although, optimal estimation of these states is made in a single retrieval process, it is made for each of the three states





independently, meaning, for instance, that the estimation is not optimal for the $\{H_2O,\delta D\}$-pairs. The reason is that the remote sensing system's sensitivity for the humidity state is much higher than for the $\delta D$-state (and significantly higher than for the deuterium-excess-state). The problem becomes clearly visible by transforming the full averaging kernel matrix onto the $\{humidity, \delta D, d\}$-proxy basis:

$$\mathbf{A}^{1'} = \mathbf{P}\mathbf{A}^{1}\mathbf{P}^{-1} \tag{18}$$

Here $\mathbf{A}^{1'}$ is the full averaging kernel matrix in the $\{humidity, \delta D, d\}$-proxy basis. It is depicted in Fig. 5, which clearly reveals larger sensitivity for humidity (kernel block $\mathbf{A}^{1}_{11}{}'$) than for $\delta D$ (kernel block $\mathbf{A}^{1}_{22}{}'$). Data with this characteristic cannot be used in the context of $\{H_2O,\delta D\}$-pair distribution analyses.

A transformation of the error covariance matrices onto the $\{humidity, \delta D, d\}$-proxy basis (transformation operation see Eq. 7) is very helpful for analysing the error characteristics of the $H_2O$, $\delta D$, and deuterium-excess data, because error covariances expressed in the $\{humidity, \delta D, d\}$-proxy basis are good proxies for the error covariances of $H_2O$, $\delta D$, and deuterium-excess (for more details see discusion in Sect. 4.2 of Schneider et al., 2012).

### 4.4.1 Aposteriori processing for a quasi optimal estimation of {H₂O,δD}-pairs

During MUSICA, an aposteriori processing method for obtaining a quasi optimal estimation product of $\{H_2O,\delta D\}$-pairs has been developed. The aposteriori processing brings about a moderate reduction of the $H_2O$ sensitivity and of the $\delta D$ cross dependencies on $H_2O$ (Sect. 4.2 in Schneider et al., 2012). The operation has to be performed with logarithmic scale full state vector, averaging kernel matrix, and covariance matrices ($x^l$, $\mathbf{A}^l$ and $\mathbf{S}^l$):

$$x^{l*} = \mathbf{P}^{-1}\mathbf{C}\mathbf{P}(x^l - x^l_a) + x^l_a, \tag{19}$$

$$\mathbf{A}^{l*} = \mathbf{P}^{-1}\mathbf{C}\mathbf{P}\mathbf{A}^l, \text{ and} \tag{20}$$

$$\mathbf{S}^{l*} = \mathbf{P}^{-1}\mathbf{C}\mathbf{P}\mathbf{S}^l\mathbf{P}^T\mathbf{C}^T\mathbf{P}^{-T}. \tag{21}$$

Here, $x^{l*}$, $\mathbf{A}^{l*}$, and $\mathbf{S}^{l*}$ are aposteriori processed logarithmic scale full state vector, averaging kernel matrix, and covariance matrices, respectively. Operator $\mathbf{P}$ is introduced by Eq. (5) and the aposteriori operator $\mathbf{C}$ is:

$$\mathbf{C} = \begin{pmatrix} \mathbf{A}^{1}_{22}{}' & 0 & 0 \\ -\mathbf{A}^{1}_{21}{}' & \mathbb{I} & 0 \\ 0 & 0 & \mathbb{I} \end{pmatrix}. \tag{22}$$

Figure 6 depicts the aposteriori processed averaging kernel $\mathbf{A}^{l*}$. It is obvious that the processing ensures that the sensitivity for humidity (kernel block $\mathbf{A}^{l*}_{11}$) and for $\delta D$ (kernel block $\mathbf{A}^{l*}_{22}$) are almost identical. Data with this characteristic are provided on a linear scale in the "ftir.iso.post.h2o" HDF files and they are well-suited for $\{H_2O,\delta D\}$-pair distribution analyses.



### 4.4.2 Aposteriori processing for a quasi optimal estimation of {H₂O,δD,d}-triplets

Fig. 6 reveals that kernel block $\mathbf{A}_{33}^{l*}$ is still rather different from the kernel blocks $\mathbf{A}_{11}^{l*}$ and $\mathbf{A}_{22}^{l*}$. Furthermore, there is a rather large cross dependency of $\delta D$ on deuterium-excess (kernel block $\mathbf{A}_{32}^{l*}$). This means that aposteriori correction with operator $\mathbf{C}$ according to Eq. (22) is not sufficient for providing deuterium excess data that can be analysed together with $H_2O$ and $\delta D$ and that is of sufficient quality. For such purpose, the aposteriori treatment has to be stronger, which can be achieved by using the following operator $\mathbf{C}$:

$$\mathbf{C} = \begin{pmatrix} \mathbf{A}_{33}^{l\,'} & 0 & 0 \\ -\mathbf{A}_{21}^{l\,'} & \mathbf{A}_{33}^{l\,'} & 0 \\ -\mathbf{A}_{31}^{l\,'} & -\mathbf{A}_{32}^{l\,'} & \mathbb{I} \end{pmatrix}. \tag{23}$$

Figure 7 depicts the aposteriori processed averaging kernel $\mathbf{A}^{l*}$ by using $\mathbf{C}$ from Eq. (23). This treatment ensures that the three blocks $\mathbf{A}_{11}^{l*}$, $\mathbf{A}_{22}^{l*}$ and $\mathbf{A}_{33}^{l*}$ are almost identical. Nevertheless, the retrieved deuterium-excess shows still some cross dependency on $\delta D$ (averaging kernel block $\mathbf{A}_{32}^{l*}$). Currently, these data are not provided via the NDACC database, mainly because it has so far not been possible to empirically prove the quality of this deuterium-excess remote sensing data. In any case, interested users are welcome to investigate the quality of these data. The required aposteriori processing can be made by using the data provided in "ftir.iso.h2o" HDF files and by following the description given in this paper (in case of doubts, please contact the MUSICA team).

## 5 Conclusions

For a correct optimal estimation retrieval of the full water vapour isotopologue state, we have to consider that atmospheric variations of different isotopologues are strongly correlated. This strong correlation is then also present in the retrieved state vectors and it has to be considered when interpreting averaging kernels and error covariances. As a consequence, it makes little sense to provide different isotopologues in the form of individual states and via individual datasets. Instead, it is essential that water vapour isotopologues are made available as single full state vectors together with their full averaging kernels and error covariances. The standard GEOMS metadata template for FTIR data on the NDACC database (called GEOMS-TE-FTIR) does not allow data to be provided in such a format and a slight extension of the standard FTIR template has been made (the modified template is called GEOMS-TE-FTIR-ISO-001). The MUSICA NDACC/FTIR data are now available in this new data format on the NDACC database and via doi:10.5281/zenodo.48902. The extended template can also be used for providing isotopologue data of other molecules (e.g. ozone isotopologue data).

In order to be compliant with GEOMS, the data are provided on a linear scale and as volume mixing ratio (ppmv). However, since the retrieval is performed on a logarithmic scale, it is recommended to transfer states, kernels, and averaging kernels onto a logarithmic scale. On this scale, linearity in the context of averaging kernel operations can be assumed and we can furthermore make transformations between the $\{\ln[H_2^{16}O], \ln[H_2^{18}O], \ln[HD^{16}O]\}$-basis and the $\{\mathrm{humidity}, \delta D, d\}$-proxy basis. The transformation of state vector, error covariances and averaging kernels onto the $\{\mathrm{humidity}, \delta D, d\}$-proxy basis gives





insight into the actual sensitivity and error characteristics of the retrieved humidity, $\delta$D, and deuterium-excess values. It can be shown that {$H_2O$,$\delta$D}-data pairs can be obtained in an optimal estimation sense, but only if the retrieval output undergoes an aposteriori processing.

The MUSICA NDACC/FTIR data are made available in the form of two different data types. The first type ("ftir.iso.h2o")

is the direct retrieval output. It reports the optimal estimations of the states $\ln[H_2^{16}O]$, $\ln[H_2^{18}O]$ and $\ln[HD^{16}O]$ and a user should work with these data for studies that focus on water vapour and disregard the difference between the isotopologues. The second type ("ftir.iso.post.h2o") is the aposteriori processed output. It reports the optimal estimation of {$H_2O$,$\delta$D}-pairs and it should be used for analysing moisture pathways by means of {$H_2O$,$\delta$D}-distribution plots. Table 3 summarises the guidelines that should be considered for a correct data usage.

**Appendix A:  Modification of HITRAN 2012 line parameter**

For the ground-based FTIR retrieval, consideration of a non-Voigt line shape parameterisation becomes important, because of the very high resolution spectra (Full Width Half Maximum of the instrumental line shape of about $0.005\,\mathrm{cm}^{-1}$).

We allow for a speed-dependent Voigt line shape, whereby we assume a $\Gamma_2/\Gamma_0$ of 15%, which is in good agreement with previous studies (e.g. D'Eu et al., 2002; Schneider et al., 2011) and fit line intensity ($S$) and pressure broadening ($\gamma_{\mathrm{air}}$). We use

six high quality $H_2O$ and $\delta$D in-situ profiles measured during an aircraft campaign in the surroundings of Tenerife Island and in coincidence with ground-based FTIR observations (Dyroff et al., 2015; Schneider et al., 2015) for empirically estimating the overall errors in $S$ and $\gamma_{\mathrm{air}}$. In addition, we use FTIR spectra measured at three distinct sites (Izaña, Karlsruhe, and Kiruna) for eliminating inconsistencies between the parameters of the different lines. Theory, practise, and limitations of such empirical line parameter optimisation method are discussed in Schneider and Hase (2009) and Schneider et al. (2011).

Table 4 resumes the modifications we had to make on the HITRAN 2012 parameters for different lines of Fig. 1 in order to adjust them for a speed-dependent Voigt line shape and for bringing them into agreement with coincident ISOWAT profile measurements and for minimising the residuals in the spectral fits at Izaña, Karlsruhe, and Kiruna. The obtained values are in agreement with our previous studies (Schneider and Hase, 2009; Schneider et al., 2011) and they are reasonable in the sense that they lie within the uncertainty ranges as given in the HITRAN data files. A value for $\Gamma_2/\Gamma_0$ of 15% means a line narrowing,

which in a Voigt line shape model could be approximated by reducing $\gamma_{\mathrm{air}}$ by 4%. In order to counterbalance, parameter $\gamma_{\mathrm{air}}$ had to be generally increased (see last column in Table 4).

*Acknowledgements.* We would like to thank the many different technicians, PhD students, post-docs, and scientists from the different research groups that have been involved in the NDACC-FTIR activities during the last two decades. Thanks to their excellent work (maintenance, calibration, observation activities, etc.) high quality long-term datasets can be generated.

The Eureka measurements were made at the Polar Environment Atmospheric Research Laboratory (PEARL) by the Canadian Network for the Detection of Atmospheric Change (CANDAC), led by James R. Drummond, and in part by the Canadian Arctic ACE Validation Campaigns, led by Kaley A. Walker. They were supported by the AIF/NSRIT, CFI, CFCAS, CSA, EC, GOC-IPY, NSERC, NSTP, OIT,





PCSP, and ORF. The authors wish to thank PEARL site manager Pierre F. Fogal, the CANDAC operators, and the staff at Environment Canada's Eureka weather station for their contributions to data acquisition, and logistical and on-site support.

We thank the Alfred Wegener Institut Bremerhaven for support in using the AWIPEV research base, Spitsbergen, Norway. The work has been supported by EU-Project NORS.

We would like to thank Uwe Raffalski and Peter Völger for technical support at IRF Kiruna.

The University of Liège contribution to the present work has primarily been supported by the A3C PRODEX program, funded by the Belgian Federal Science Policy Office (BELSPO, Brussels), and by the Swiss GAW-CH program of MeteoSwiss (Zürich). Laboratory developments and mission expenses were funded by FRS-FNRS and the Fédération Wallonie-Bruxelles, respectively. We thank the International Foundation High Altitude Research Stations Jungfraujoch and Gornergrat (HFSJG, Bern) for supporting the facilities needed to perform the

observations.

E. Sepúlveda is supported by the Ministerio de Economía y Competitividad from Spain under the project CGL2012-37505 (NOVIA project).

The measurements in Mexico (Altzomoni) are supported by UNAM-DGAPA grants (IN109914, IN112216) and Conacyt (239618, 249374). Start-up of the measurements in Altzomoni was supported by International Bureau of BMBF under contract no. 01DN12064.

Special thanks to A. Bezanilla for data management and the RUOA program (www.ruoa.unam.mx) and personnel for helping maintaining the station.

Measurements at Wollongong are supported by the Australian Research Council, grant DP110103118.

We would like to thank Antarctica New Zealand and the Scott Base staff for providing logistical support for the NDACC-FTIR measurement program at Arrival Heights.

This study has been conducted in the framework of the project MUSICA which is funded by the European Research Council under the European Community's Seventh Framework Programme (FP7/2007-2013) / ERC Grant agreement number 256961.

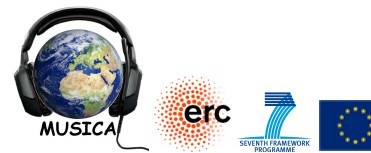



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





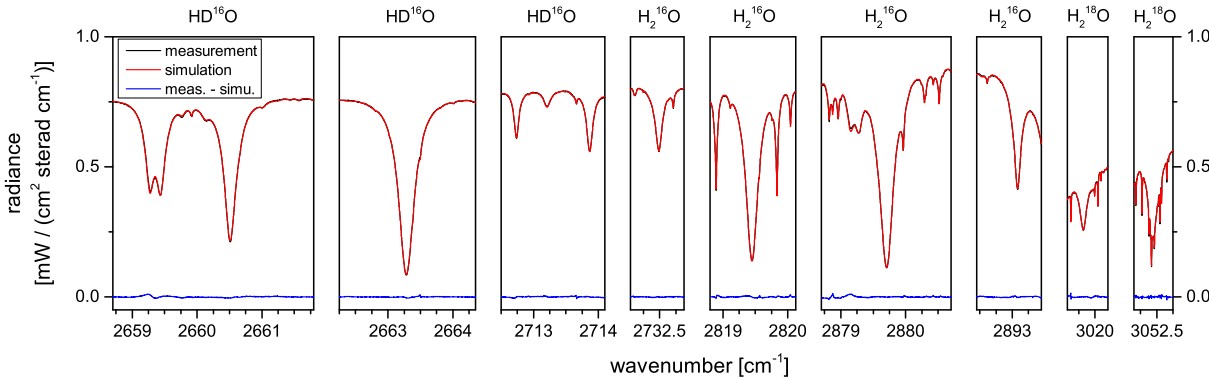

**Figure 1.** The spectral windows used for the MUSICA ground-based NDACC/FTIR retrievals. Shown is an example for a typical measurement (Karlsruhe 15 September 2011, 12:03 UT; solar elevation: 43.1°; $H_2O$ slant column: 22.9 mm). Black line: measurement; red line: simulation; blue line: residual (difference between measurement and simulation).





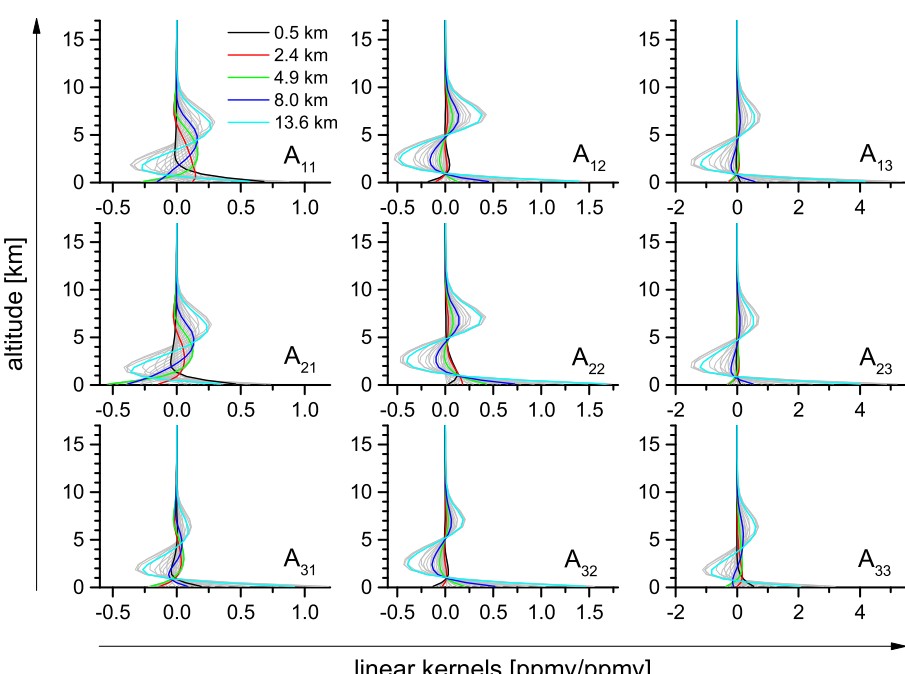

**Figure 2.** Column entries of the nine blocks of the full averaging kernel matrix **A** (see Eq. 10). This kernel is for the retrieval with the spectrum and fit as shown in Fig. 1.





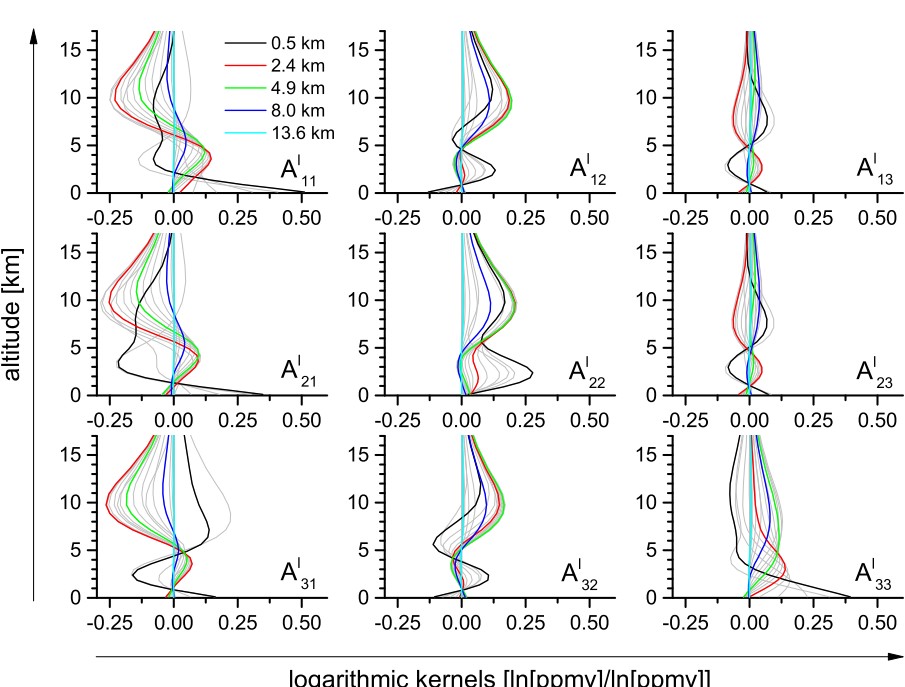

**Figure 3.** Same as Fig. 2, but for the logarithmic scale kernel matrix $\mathbf{A}^1$.





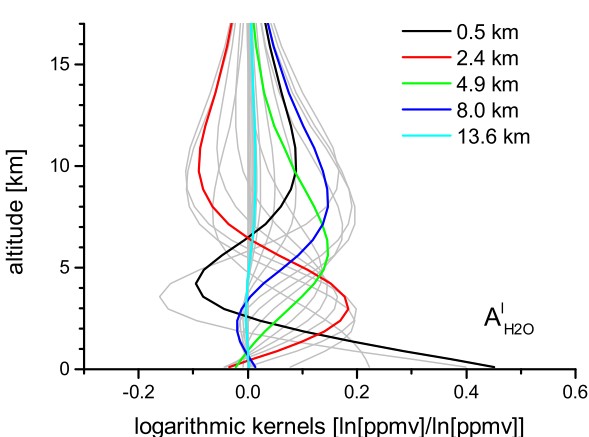

**Figure 4.** Logarithmic scale kernel matrix $\mathbf{A_{H2O}^l} = \mathbf{A_{11}^l} + \mathbf{A_{12}^l} + \mathbf{A_{13}^l}$, with the $\mathbf{A^l}$ blocks as plotted in Fig. 3.



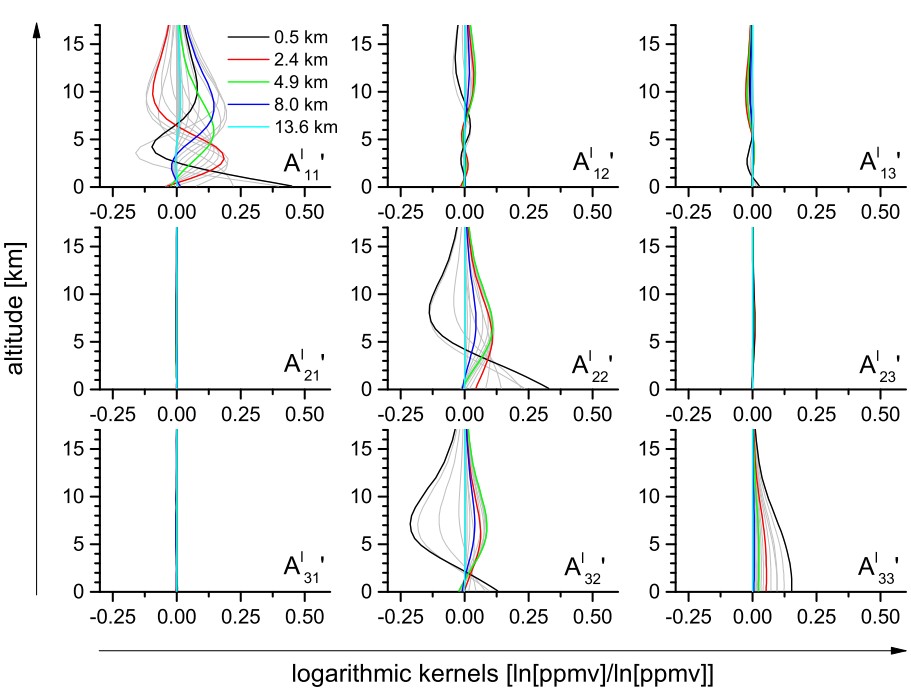

**Figure 5.** Column entries of the nine blocks of the logarithmic scale kernel matrix in the {humidity, $\delta$D, d}-proxy state ($\mathbf{A}^{1\prime}$, see Eq. 18).





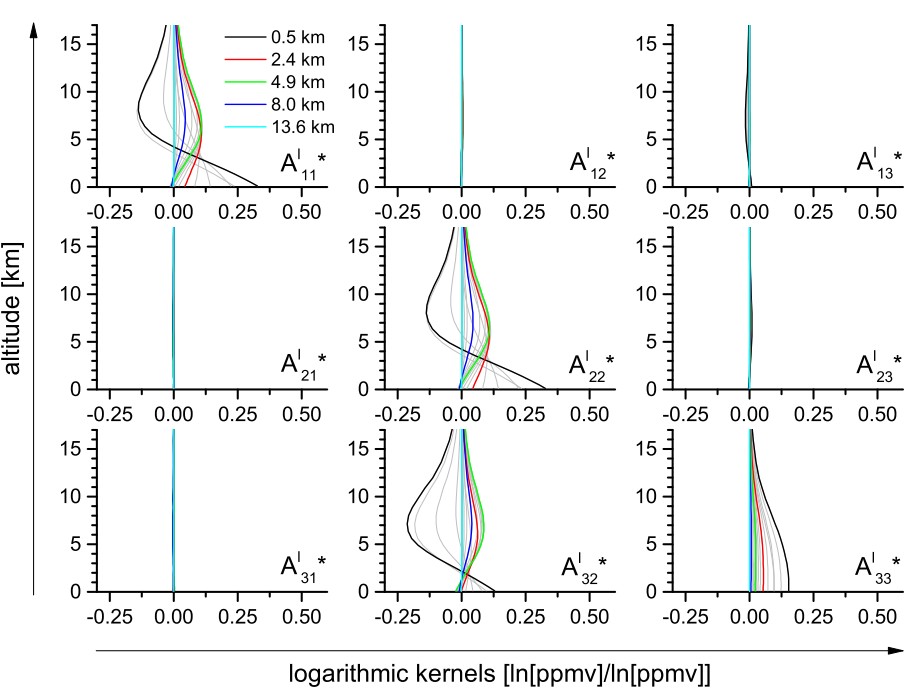

**Figure 6.** Same as Fig. 5, but after aposteriori processing with **C** from Eq. 22 ($\mathbf{A}^{1*}$, see Eq. 20).





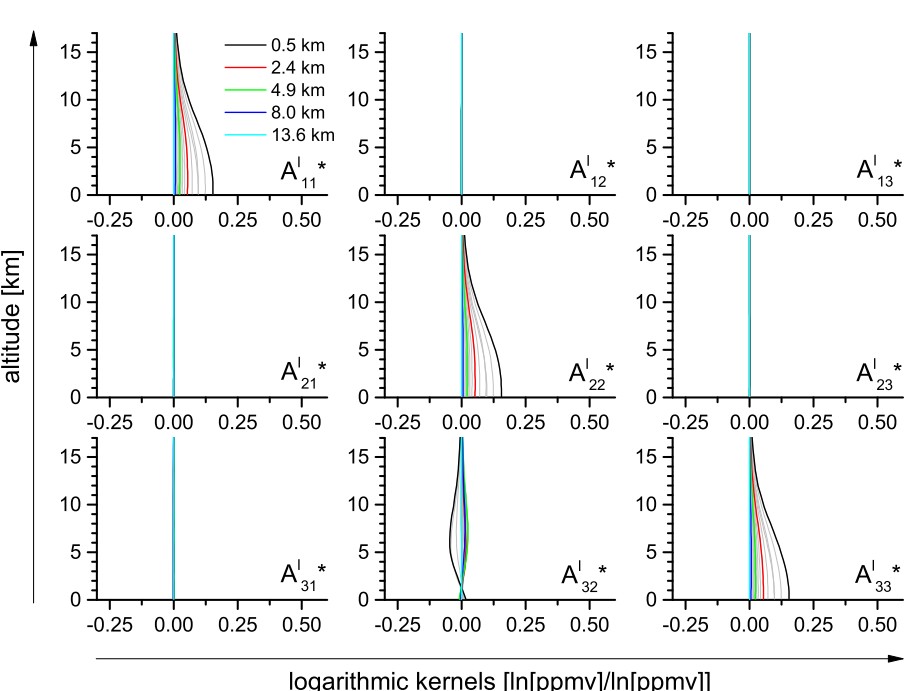

**Figure 7.** Same as Fig. 6, but using **C** from Eq. 23.



**Table 1.** List of current MUSICA NDACC/FTIR sites (ordered from North to South) and available MUSICA data record. DOFS (Type 1) report the typical trace of $\mathbf{A^1_{H2O}}$ and DOFS (Type 2) of $\mathbf{A^{1*}_{11}} \approx \mathbf{A^{1*}_{22}}$ for optimal estimation of H2O-deltaD pairs (example kernel plotted in Fig. 5).

| Site | Location | Altitude | Data Record | DOFS (Type 1) | DOFS (Type 2) |
|------|----------|----------|-------------|---------------|---------------|
| Eureka, Canada | 80.1°N, 86.4°W | 610 m a.s.l. | 2006-2014 | 2.9 | 1.7 |
| Ny Alesund, Norway | 78.9°N, 11.9°E | 21 m a.s.l. | 2005-2014 | 2.8 | 1.6 |
| Kiruna, Sweden | 67.8°N, 20.4°E | 419 m a.s.l. | 1996-2014 | 2.8 | 1.6 |
| Bremen, Germany | 53.1°N, 8.9°E | 27 m a.s.l. | 2004-2014 | 2.8 | 1.6 |
| Karlsruhe, Germany | 49.1°N, 8.4°E | 110 m a.s.l. | 2010-2014 | 2.8 | 1.6 |
| Jungfraujoch, Switzerland | 46.6°N, 8.0°E | 3580 m a.s.l. | 1996-2014 | 2.7 | 1.6 |
| Izaña/Tenerife, Spain | 28.3°N, 16.5°W | 2367 m a.s.l. | 1999-2014 | 2.9 | 1.7 |
| Altzomoni, Mexico | 19.1°N, 98.7°W | 3985 m a.s.l. | 2012-2014 | 2.7 | 1.7 |
| Addis Ababa, Ethiopia | 9.0°N, 38.8°E | 2443 m a.s.l. | 2009-2013 | 2.6 | 1.6 |
| Wollongong, Australia | 34.5°S, 150.9°E | 30 m a.s.l. | 2007-2014 | 2.7 | 1.6 |
| Lauder, New Zealand | 45.1°S, 169.7°E | 370 m a.s.l. | 1997-2014 | 2.8 | 1.6 |
| Arrival Heights, Antarctica | 77.8°S, 166.7°E | 250 m a.s.l. | 2002-2014 | 2.7 | 1.4 |

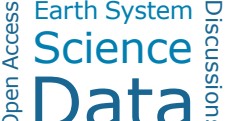



**Table 2.** Uncertainty sources used for the error estimation. The second column gives the uncertainty value and the third column the partitioning between statistical and systematic sources.

| Error Source | Uncertainty | Statistical/Systematic |
|---|---|---|
| Measurement Noise | 0.4% | 100/0 |
| Baseline (Channeling, assuming 4 frequencies: 0.005, 0.2, 1.0, and 3.0 cm$^{-1}$) | 0.2% | 50/50 |
| Baseline (Offset) | 0.1% | 50/50 |
| Instrumental Line Shape (Mod. Eff. and Pha. Err.) | 10% and 0.1 rad | 50/50 |
| Temperature Profile | 3 K (surface - 5 km a.s.l.) | 70/30 |
| | 3 K (5 - 12 km a.s.l.) | 70/30 |
| | 3 K (above 12 km a.s.l.) | 70/30 |
| Line Of Sight | 0.1° | 90/10 |
| Solar Lines (Intensity and $\nu$-scale) | 1% and $10^{-6}$ | 80/20 |
| Spectroscopic Parameters ($S$ and $\gamma$) | +1% ($H_2^{16}O$) | 0/100 |
| | -1% ($H_2^{18}O$) | 0/100 |
| | +2% ($HD^{16}O$) | 0/100 |





**Table 3.** Summary of recommendations and comments for the two principal types of data users.

| User Type | HDF File | Recommendations and Comments |
|---|---|---|
| H$_2$O profiles | ftir.iso.h2o | transfer data onto a logarithmic scale (see Sect. 4.1) |
| | | use H$_2^{16}$O retrieval products as H$_2$O (more than 99.7% of all water vapour is in form of H$_2^{16}$O) |
| | | use averaging kernel $\mathbf{A^1_{H2O} = A^1_{11} + A^1_{12} + A^1_{13}}$ (see Sect. 4.3) |
| {H$_2$O,$\delta$D}-pairs | ftir.iso.post.h2o | comment: all "ftir.iso.post.h2o" retrieval products are corrected as described in Sect. 4.4.1 |
| | | transfer data onto a logarithmic scale (see Sect. 4.1) |
| | | use data in the $\{\mathrm{humidity}, \delta\mathrm{D}, \mathrm{d}\}$-proxy basis for sensitivity and error assessments (see Sect. 4.4) |

**Table 4.** Modifications in the line parameters (line intensity and pressure broadening) made with respect to HITRAN 2012.

| Line Centre [cm$^{-1}$] | Isotopologue | $\Delta$S [%] | $\Delta\gamma$ [%] |
|---|---|---|---|
| 2660.511700 | HD$^{16}$O | -5.52 | +3.96 |
| 2663.285820 | HD$^{16}$O | -5.53 | +4.00 |
| 2713.862650 | HD$^{16}$O | -5.53 | +4.07 |
| 2732.493160 | H$_2^{16}$O | +12.26 | +9.35 |
| 2819.449040 | H$_2^{16}$O | -3.07 | +4.52 |
| 2879.706660 | H$_2^{16}$O | -8.26 | +6.84 |
| 2893.075920 | H$_2^{16}$O | -9.07 | +9.64 |
| 3019.824500 | H$_2^{18}$O | -5.40 | -0.72 |
| 3052.444870 | H$_2^{18}$O | -6.32 | -0.71 |