# Peer review of "Tropospheric water vapour isotopologue data ($H_2^{16}O$, $H_2^{18}O$ and $HD^{16}O$) as obtained from NDACC/FTIR solar absorption spectra"

_Earth System Science Data, 2016_

## Referee Comment (RC1) · Anonymous Referee #1 · 8 Jul 2016

The manuscript "Tropospheric water vapour isotopologue data (H216O, H218O and HD16O) as obtained from NDACC/FTIR solar absorption spectra" presents the technique for retrieving water vapor isotopologue measurements from FTIR data. It's a well-developed framework at this point, and a very important one, so I think the point of this manuscript is to serve as a basic reference for the technique. I am primarily a user of the data rather than a connoisseur of retrieval techniques, so I hope that someone more well-versed in the mathematics can check over the details. My only concern about the manuscript is that the mathematical framework is presented without a lot of hand-holding or examples that can help a non-specialist to understand the technique. This has been a general problem I have had with remote sensing papers - I really

want to understand the underlying techniques, but the papers are presented in a fairly abstract way with few meaningful examples. It's almost certainly my own fault rather than the authors, but I wonder if they could add some more descriptive text and some examples that can help the lay reader better make the connections between the mathematical foundations and the ultimate applications to atmospheric sciences. It would increase the impact of the manuscript if the paper were more accessible to a broader audience. That may not really be the goal of the paper, which is OK, but I would love for some group (and this group of authors is as well-qualified as any) to write a tutorial paper that can help someone like myself better understand how to use FTIR and other remote sensing data.

Overall, this paper represents an important development in our ability to use FTIR datasets. I would hope the authors can add some small amount of additional material to make it more accessible, but basically I think this is a fine manuscript.
* * *

---

## Referee Comment (RC2) · Anonymous Referee #2 · 3 Aug 2016

Reviewer #2

Manuscript essd-2016-9:

"Tropospheric water vapour isotopologue data (H216 O, H218O and HD16O) as obtained from NDACC/FTIR solar absorption spectra"

Authors: Sabine Barthlott et al.

Overall impression: The manuscript is of high quality and relevance and I would recommend publications after some very minor reviews.

The methods and mathematical details given are sound. Some of the covariance ma-

trix operations are complex, almost intimidating, but also elegant in describing the data in the best possible way and according to intended use (total water vapour or isotopologue signal). The references to other and previous work appear to be quite complete and appropriate.

All data are of high quality and are readily accessible, both via the NDACC portal and through the doi provided. A thorough and convincing quantitative discussion of systematic and random errors is presented. Even details of the metadata description are given, which often find little attention in other work. The observation methods, instruments and data processing are all state of the art and the data processing methods are truly cutting edge that I hope others will take on board. The paper is also making a clear stand for standards. This includes instrument calibration, metadata, observational data and a complete error analysis that can be traced back to SI standards in an unbroken chain. The data is optimally suited for long-term climate data record analysis and the validation of shorter term observations including satellites.

The data presented is significant, unique, useful and complete. There are several messages that I'm taking from this manuscript that the work implies, perhaps without spelling it out too clearly, but that I feel could be highlighted in a review article of the NDACC special issue: isotopologue data needs to be treated differently from "normal" atmospheric composition measurements and the authors explain neatly why this is so. The spatial and temporal variability of water vapour in the atmosphere is extremely high making it very difficult to cross-calibrate or validate different instruments. However, the variability of isotopologuous signals vary on comparatively much larger scales thus offering a better posed alternative for validation work. Models are heavily underutilising isotopologue data, likely due to the fact that most climate and NWP models are still not capable of handling isotopologues including their pathways. Modellers need to seriously catch up in this field. Isotopologue data of in particular water vapour has the potential to give detailed and new insights into moisture pathways that could significantly improve climate models including paleo models. This work describes the

best practise for ground-based FTIR observations for water vapour including its iso-topologues, but it also provides a formalism that other instruments from ground, air or space could easily adopt.

I could find no inconsistencies in the data presented. The discussion is plausible and concise.

The presentation is of high quality, appropriate length, well structured and the authors have good command of the English language. The use of maths and symbols is consistent with modern science and tables and figures are appropriate.

I'm confident that scientists working in this field are able to understand and use the data and its methods as described.

Rating of this manuscript: Significance: the uniqueness, usefulness and completeness warrant a rating of Excellent in my humble opinion. Data from all over the globe have been processed in a very consistent and innovative way, keeping applicable and defining new standards as adequate Data quality: Again I opt for Excellent. I have worked with NDACC data in the past and in particular the Karlsruhe group has never been short of the best possible practice, always trying to push the borders. Presentation quality: Very good. Some of the maths is complex and I'm uncertain whether a bit more explanation would have been helpful or whether it is my lack of knowledge or un-familiarity with some aspects of it. There are a few minor comments on the text detailed below that I would ask the authors to consider before going to print.

I wish to congratulate the authors on their sound work and thank the many people that made this contribution to science possible through collecting and analysing the observations presented.

Below I'm listing a number of specific comments and suggestions that I would ask the authors to consider (p.=page, L=Line number):

p2L2: "deuterium-excess (d = $\delta$D $-$ 8$\delta$ 18 " I was immediately curious where the factor

8 came from or whether it might be a typo – a short comment either here or later in the discussion where it appears again would be really nice. Inclined readers can then still read up the details in the reference provided, but a simple explanation would have most readers reading on happily.

P2L3: 'are' not 'is' "particularly promising."

p2L13: "since the 1990s, " - some data records at sites like Jungfraujoch and Kitt Peak are much longer and could be of much interest for long term climate studies or is the quality of pre-1990s data too poor?

p2L13: " are of particular interest " 'to' not 'for'

p3L15: " vary mostly in" 'parallel': I think alike, uniformly or in equal measure would be the better word here instead of parallel

p3L18: " is about two orders of magnitude smaller than the correlated variations of ln [H16 2 O], ln [H2 18 O] and ln [HD16 O]." It would be nice to add a reference here.

P3L19/20: "(due to the log-normal distribution characteristic)" Is this ultimately due to the logarithmic drop in pressure and water vapour VMR with altitude?

P3, definitions/equations 2, 3, 4: I would write equations; i.e. add hum=, $\delta$D= and d= to the left of your definitions for easier reading – in particular when flicking back to them when chewing through the maths on subsequent pages.

P5L16: "All these spectral windows are" consider replacing "covered by" with 'observed within/recorded with the same optical' "NDACC filter #3."

p8L9: " the retrieved state vector element i" replace 'responses" with "responds"

p9 eq(14): does this come from SMOW? See also comment above on factor 8

p12L3: Comment: would the use of more microwindows help to reduce uncertainties? Or are current spectroscopy parameters to inconsistent/poor? Or is it inadequacies in

the line shape model used? Or a combination of the latter? I think a sentence on how the accuracy could be pushed further would be well placed here, independent of how difficult it may be to achieve that.

Figure 2: Please explain all figure captions, e.g. what does "0.5 km" mean in this context. Make clear if random or systematic errors are illustrated. Should some of it be too long, move some of it into the text but make that clear connection between figure and what is being shown.

Figure 7: Does this mean that one trades profile or vertical information for isotopologue information? This would then also illustrate why one state parameter representation is more adequate than another for reporting overall humidity as opposed to isotopologue ratios.

P24, Table 2: Consider writing "Modulation Efficiency and Phase Error", no need to abbreviate as not all readers may be familiar with the abbreviations.

P24, Table 2: Why separate the Temperature profile into different altitude slices if the uncertainties are the same for all 3 slices?

Please also note the supplement to this comment:
http://www.earth-syst-sci-data-discuss.net/essd-2016-9/essd-2016-9-RC2-supplement.pdf

---

## Author Comment (AC1) · 27 Sep 2016

Dear referees,

Thank you very much for your effort in evaluating, commenting, and thus improving our manuscript and for the time spent to perform this work.
We have elaborated our replies and hope that they are satisfying for you.

Best regards,
Sabine Barthlott

(Referees' text in *italic*, authors' reply indented and blue)

*Anonymous Referee #1*

*The manuscript "Tropospheric water vapour isotopologue data (H216O, H218O and HD16O) as obtained from NDACC/FTIR solar absorption spectra" presents the technique for retrieving water vapor isotopologue measurements from FTIR data. It's a well-developed framework at this point, and a very important one, so I think the point of this manuscript is to serve as a basic reference for the technique.*

> The main objective of our paper is to explain the published MUSICA dataset (which is similar to other NDACC/FTIR datasets), to give recommendations for different applications and with it to assure the correct usage of our data.

*I am primarily a
user of the data rather than a connoisseur of retrieval techniques, so I hope that someone more well-versed in the mathematics can check over the details. My only concern about the manuscript is that the mathematical framework is presented without a lot of hand-holding or examples that can help a non-specialist to understand the technique. This has been a general problem I have had with remote sensing papers - I really want to understand the underlying techniques, but the papers are presented in a fairly abstract way with few meaningful examples. It's almost certainly my own fault rather than the authors, but I wonder if they could add some more descriptive text and some examples that can help the lay reader better make the connections between the mathematical foundations and the ultimate applications to atmospheric sciences. It would increase the impact of the manuscript if the paper were more accessible to a broader audience. That may not really be the goal of the paper, which is OK, but I would love for some group (and this group of authors is as well-qualified as any) to write a tutorial paper that can help someone like myself better understand how to use FTIR and other remote sensing data.*
*Overall, this paper represents an important development in our ability to use FTIR datasets. I would hope the authors can add some small amount of additional material to make it more accessible, but basically I think this is a fine manuscript.*

> We see the point of Refereee #1. We'll try to shortly summarize some main basics in an additional chapter in the appendix and to explain some essential definitions to ease the understanding also for people that are not so familiar with remote sensing data.

*Reviewer #2*

*Overall impression:*
*The manuscript is of high quality and relevance and I would recommend publications after some*
*very minor reviews.*
*The methods and mathematical details given are sound. Some of the covariance matrix operations are complex, almost intimidating, but also elegant in describing the data in the best possible way and according to intended use (total water vapour or isotopologue signal). The references to other and previous work appear to be quite complete and appropriate.*
*All data are of high quality and are readily accessible, both via the NDACC portal and through the*
*doi provided. A thorough and convincing quantitative discussion of systematic and random errors is presented. Even details of the metadata description are given, which often find little attention in other work. The observation methods, instruments and data processing are all state of the art and the data processing methods are truly cutting edge that I hope others will take on board. The paper is also making a clear stand for standards. This includes instrument calibration, metadata, observational data and a complete error analysis that can be traced back to SI standards in an unbroken chain. The data is optimally suited for long-term climate data record analysis and the validation of shorter term observations including satellites.*

*The data presented is significant, unique, useful and complete. There are several messages that I'm taking from this manuscript that the work implies, perhaps without spelling it out too clearly, but that I feel could be highlighted in a review article of the NDACC special issue:*

- *isotopologue data needs to be treated differently from "normal" atmospheric composition measurements and the authors explain neatly why this is so.*
- *The spatial and temporal variability of water vapour in the atmosphere is extremely high making it very difficult to cross-calibrate or validate different instruments. However, the variability of isotopologuous signals vary on comparatively much larger scales thus offering a better posed alternative for validation work.*
- *Models are heavily underutilising isotopologue data, likely due to the fact that most climate and NWP models are still not capable of handling isotopologues including their pathways. Modellers need to seriously catch up in this field.*
- *Isotopologue data of in particular water vapour has the potential to give detailed and new insights into moisture pathways that could significantly improve climate models including paleo models.*
- *This work describes the best practise for ground-based FTIR observations for water vapour including its isotopologues, but it also provides a formalism that other instruments from ground, air or space could easily adopt.*

*I could find no inconsistencies in the data presented. The discussion is plausible and concise. The presentation is of high quality, appropriate length, well structured and the authors have good*
*command of the English language. The use of maths and symbols is consistent with modern science and tables and figures are appropriate.*
*I'm confident that scientists working in this field are able to understand and use the data and its*
*methods as described.*

*Rating of this manuscript:*

*Significance: the uniqueness, usefulness and completeness warrant a rating of Excellent in my humble opinion. Data from all over the globe have been processed in a very consistent and*
*innovative way, keeping applicable and defining new standards as adequate*

*Data quality:*
*Again I opt for Excellent. I have worked with NDACC data in the past and in particular the Karlsruhe group has never been short of the best possible practice, always trying to push the borders.*

*Presentation quality: Very good. Some of the maths is complex and I'm uncertain whether a bit more explanation would have been helpful or whether it is my lack of knowledge or unfamiliarity with some aspects of it. There are a few minor comments on the text detailed below that I would ask the authors to consider before going to print.*

*I wish to congratulate the authors on their sound work and thank the many people that made this contribution to science possible through collecting and analysing the observations presented.*

> We feel honoured by this acclaim expressed by the referee. It is a pleasure for us to know that your very kind appreciation not only targets at the publication itself but also perceives the substantial work it is built upon. As the MUSICA activities are currently severely jeopardized by expiring funding, your statement is invaluable support for us at the right time.

*Below I'm listing a number of specific comments and suggestions that I would ask the authors to consider (p.=page, L=Line number):*

- *p2L2: "deuterium-excess ($d = \delta D - 8\delta 18$ " I was immediately curious where the factor 8 came from or whether it might be a typo – a short comment either here or later in the discussion where it appears again would be really nice. Inclined readers can then still read up the details in the reference provided, but a simple explanation would have most readers reading on happily.*

  > The formula of deuterium-excess including factor 8 is based on definitions used in studies as e.g.
  >
  > Craig, H.: Isotopic variations in meteoric waters, Science, 133, 1702-03, 1961.
  >
  > Craig, H.: Standard for Reporting Concentrations of Deuterium and Oxygen-18 in Natural Waters, Science, 133, 1833–1834, doi:10.1126/science.133.3467.1833, 1961.
  >
  > Dansgaard, W.: Stable isotopes in precipitation, Tellus, 16, 436–468, 1964.
  >
  > We'll add these references.

- *P2L3: 'are' not 'is' "particularly promising."*

  > Ok.

- *p2L13: "since the 1990s, " - some data records at sites like Jungfraujoch and Kitt Peak are much longer and could be of much interest for long term climate studies or is the quality of pre-1990s data too poor?*

  > This is right, these data would indeed be of interest and with our XCO2 filtering method (Barthlott et al., 2015), these data could be also quality checked even if they were measured with other instruments and settings. But unfortunately within our project we were not able to enlarge our dataset. This could be a task for follow-on projects.

- *p2L13: " are of particular interest " 'to' not 'for'*

  Ok.

- *p3L15: " vary mostly in" 'parallel': I think alike, uniformly or in equal measure would be the better word here instead of parallel*

  Ok, will be changed to 'in equal measure'.

- *p3L18: " is about two orders of magnitude smaller than the correlated variations of ln [H16 2 O], ln [H2 18 O] and ln [HD16 O]." It would be nice to add a reference here.*

  Tropospheric humidity has a high variability. The ratio between the HD16O and H216O isotopologues is much smaller (by about one order of magnitude smaller). Even smaller is the variation of deuterium excess (it is a further order of magnitude smaller). The reason is the ratio between HD16O and H216O, on one hand, and H218O and H216O, on the other hand, have a strong correlation. A first publication where this has been shown is Craig and Gordon, (1965):

  Craig, H. and Gordon, L. I.: Deuterium and oxygen 18 variations in the ocean and marine atmosphere, in proc.: Stable Isotopes in Oceanographic Studies and Paleotemperatures, Spoleto, Italy, edited by: Tongiogi, E., V. Lishi e F., Pisa, Italy, 9−130, 1965.

- *P3L19/20: "(due to the log-normal distribution characteristic)" Is this ultimately due to the logarithmic drop in pressure and water vapour VMR with altitude?*

  For constant relative humidity, water vapour concentrations can be described with a log-function with respect to temperature. To our understanding, this is the reason for the logarithmic decay of water vapour concentration with altitude and log-normal distribution of water vapour.

- *P3, definitions/equations 2, 3, 4: I would write equations; i.e. add hum=, δD= and d= to the left of your definitions for easier reading – in particular when flicking back to them when chewing through the maths on subsequent pages.*

  Ok. To be consistent with our text, we'll add 'humidity=', 'δD=', and 'd='.

- *P5L16: "All these spectral windows are" consider replacing "covered by" with 'observed within/recorded with the same optical' "NDACC filter #3."*

  Ok, will be changed to 'observed within'.

- *p8L9: " the retrieved state vector element i" replace 'responses" with "responds"*

  Ok.

- *p9 eq(14): does this come from SMOW? See also comment above on factor 8*

  No, eq14 is just a combination of eq 3 and 4, for eq15 see eq3. We'll add this in the text.

- *p12L3: Comment: would the use of more microwindows help to reduce uncertainties? Or are current spectroscopy parameters to inconsistent/poor? Or is it inadequacies in the line shape model used? Or a combination of the latter? I think a sentence on how the accuracy could be pushed further would be well placed here, independent of how difficult it may be to achieve that.*

  Error estimations are discussed in detail in Schneider et al.(2012). The random error is about 2 % for $H_2O$ and 25 ‰ for $\delta D$, whereby the leading error source is uncertainty in the atmospheric temperature profiles and artefacts in the spectral baseline (like channelling or offset). Systematic errors are dominated by uncertainties in the spectroscopic parameters. There are inconsistencies between parameters for different lines as well as inadequacies in the used line

shape model. Currently, we are not able to say what is more important. Further dedicated research projects would be needed.
We'll add a sentence in the text.

- *Figure 2: Please explain all figure captions, e.g. what does "0.5 km" mean in this context. Make clear if random or systematic errors are illustrated. Should some of it be too long, move some of it into the text but make that clear connection between figure and what is being shown.*

  Please note, that Figure 2 shows some columns of the Averaging Kernel matrix, so no errors are shown in this figure. For more details concerning error estimations, please refer to Schneider et al. (2012). We'll add an explanation of the figure captions.

- *Figure 7: Does this mean that one trades profile or vertical information for isotopologue information? This would then also illustrate why one state parameter representation is more adequate than another for reporting overall humidity as opposed to isotopologue ratios.*

  If you are only interested in humidity distribution, it is sufficient to look on H216O. More than 99.7% of all atmospheric water vapour is in form of H216O. For H216O the retrieval offers profiles.

  If you are interested in processes that control humidity, the isotopologue ratios can help a lot. Especially if you look on the relation between humidity (i.e. mainly H216O) and the ratio δD. The distribution of the {H2O,δD} pairs gives insight into transportation, evaporation, cloud prcesses, etc. However, it is of course important that you really work with {H2O,δD} pairs, meaning that the H2O and the δD data are representative for the same tropospheric altitude region. Since δD represents an average of broad tropospheric layers, we also have to smooth the H2O profiles so that they become representative for the same broad layers. Only then {H2O,δD} pair plots make sense.

- *P24, Table 2: Consider writing "Modulation Efficiency and Phase Error", no need to abbreviate as not all readers may be familiar with the abbreviations.*

  Ok.

- *P24, Table 2: Why separate the Temperature profile into different altitude slices if the uncertainties are the same for all 3 slices?*

  Table 2 just contains our uncertainty assumptions and not calculated errors. We assume uncorrelated uncertainties for the three different altitude slices. With our calculated full error covariance matrices we not only get information about the errors but also on how the errors between different altitudes as well as between the three different isotopologues are correlated. (we provide error covariance matrices, see section 3.2).